# The Relationship between Metabolic Syndrome and Frailty in Older People: A Systematic Review and Meta-Analysis

**DOI:** 10.3390/geriatrics7040076

**Published:** 2022-07-26

**Authors:** Hiep Huu Hoang Dao, Mason Jenner Burns, Richard Kha, Clara K. Chow, Tu Ngoc Nguyen

**Affiliations:** 1Westmead Applied Research Centre, Sydney Medical School, Faculty of Medicine and Health, The University of Sydney, Sydney, NSW 2145, Australia; hdao5419@uni.sydney.edu.au (H.H.H.D.); mason.burns@sydney.edu.au (M.J.B.); rkha2425@uni.sydney.edu.au (R.K.); clara.chow@sydney.edu.au (C.K.C.); 2Department of Cardiology, Westmead Hospital, Westmead, NSW 2145, Australia

**Keywords:** frailty, metabolic syndrome, diabetes, hypertension, obesity, dyslipidemia, older people

## Abstract

Aims: Both metabolic syndrome (MetS) and frailty are associated with increased all-cause mortality, yet the complex interplay between these two conditions has not adequately been elucidated. We aim to analyse the relationship between MetS and frailty through a systematic review of the literature with meta-analyses. Methods: A literature search was conducted via MEDLINE and EMBASE. Studies were included if validated frameworks for defining frailty and MetS (presence of at least 3 out of the five constitutive components: abdominal obesity, high fasting blood glucose, hypertension, hypertriglyceridaemia, and low high-density lipoprotein level) were utilised, in addition to the inclusion of participants aged 60 or older. Results: Eleven studies were included, all observational. All were in community-dwelling older people, 9 cross-sectional and 2 longitudinal. Most of the studies used Fried’s frailty phenotype. The prevalence of frailty ranged from 0.9% to 14.8% in population-based studies and 35.6% in the outpatient clinic setting. The prevalence of MetS was also higher in the outpatient clinic setting at 47.5%, compared to 17.5–41.0% in the community-dwelling populations. The meta-analysis of 11 studies showed that MetS was associated with an increased risk of frailty (pooled OR 1.73, 95% CI, 1.41–2.13). Conclusion: This systematic review and meta-analysis suggest that frailty was more prevalent in older people with MetS compared to older people without MetS. The study findings suggest the importance of frailty screening in older people with MetS and a distinct role of managing MetS in preventing frailty in older people.

## 1. Introduction

In the wake of urbanisation, where a sedentary lifestyle is gradually becoming the norm, metabolic syndrome (MetS) has emerged as one of the key public health issues. MetS is a condition associated with ageing [1]. MetS or historically known as Syndrome X describes a constellation of metabolic derangements, ranging from dyslipidaemia (hypertriglyceridaemia and low HDL) to central adiposity, insulin resistance and hypertension [2]. The pathogenesis of MetS includes both genetic and modifiable factors, which have a confounding effect that eventuates in the proinflammatory environment that leads to cardiovascular disease (CVD) and other MetS-related adverse events. Early intervention to target lifestyle and risk-factor modification could prevent the progression of comorbidities to key MetS related outcomes such as diabetes, CVD, chronic liver disease and dementia [3]. According to the classification framework established by the National Cholesterol Education Program (Adult’s Treatment Panel III), the definition of metabolic syndrome warrants the presence of ≥3 of the aforementioned adverse features (NCEP 2001) [4]. Prevalence of MetS increased with ageing and varies across different geographical regions. In European populations MetS prevalence increases from 11% in males aged 20–29 to 47.2% in males aged 80–89; similarly, there is an increase from 9.2% to 64.4% in females aged 20–29 and 80–89, respectively, [1]. While 11.9–37.1% of the population in the Asia-Pacific region was reported to acquire MetS [5], a systematic review of 10 large cohort studies revealed a MetS prevalence of 11.6–26.3% in European settlements [6]. This figure can be more than 40% in regions such as South America or Africa [7]. MetS has a profoundly negative impact on health as it increases the risk of vascular disorders such as stroke, coronary heart disease and, consequently, poorer quality of life [8]. The syndrome is particularly problematic within elderly populations as not only does the prevalence of MetS increase with age, but a combination of MetS and ageing is also associated with greater all-cause mortality [9,10].

Frailty also arises as a prominent issue for the elderly due to its debilitating impacts on health outcomes. Frailty is a geriatric syndrome, characterised by a gradual decline in homeostatic tolerance and physiological reserve following exposure to stressors [10,11]. Frailty predisposes older people to falls, delirium, hospitalisations or even death and thus is regarded as a crucial transition between healthy ageing and disability [10,12].

Several studies have examined the association between frailty with constitutive components of MetS, for instance, insulin resistance and diabetes. Diabetes is known to increase hospitalization, disability, and mortality in frail individuals. Although the full extent of this effect on those with frailty is not known, impairments in physical and cognitive domains have been observed [13]. In addition, diabetes and frailty are confounding risk factors for fragility fractures observed at higher rates than expected from these conditions alone [14]. However, the relationship of frailty with MetS as a whole constellation has not been thoroughly elucidated. A low-grade chronic inflammation state, high circulating inflammatory markers and neuroendocrine dysfunction were regarded as prominent pathophysiological features across both syndromes [15,16]. As MetS and frailty both hindered the elder’s quality and quantity of life in their own regard, it is important to study such relationships for appropriate preventative strategies and interventions.

Elucidation of a positive association between frailty and MetS would highlight the importance of focusing efforts on early treatment of MetS to reduce associated morbidity and mortality. Treatment of the constitutive components of MetS through drug therapy and healthy lifestyle promotion may help to reduce frailty in addition to lessening the progression of CVD and diabetes.

Additionally, this positive association would highlight a need for customised preventative measures for the population with concomitant MetS and frailty due to the significantly worse outcomes associated with both. We conducted this systematic review and meta-analysis to examine the relationship between metabolic syndrome and frailty in older people. We hypothesised that older people with metabolic syndrome have a higher risk of having frailty.

## 2. Methods

### 2.1. Search Strategy

This systematic review (not related to a registered protocol) was conducted in alignment with checklist from Preferred Reporting Items for Systematic Reviews and Meta-Analyses (PRISMA). We conducted electronic searches of the Ovid MEDLINE database for relevant papers published from MEDLINE’s inception (1946) to February 2020, utilising keywords and Medical Subject Headings (MeSH) associated with Metabolic syndrome, Syndrome X, Insulin Resistance, elderly population and frailty. To address our research question, the search strategy was formulated in reference to PICO (population, intervention/exposure, control/comparison, and outcome) format. People of older ages were regarded as the population of interest while “metabolic syndrome” or “syndrome X” were deemed as exposure. The comparison group comprised older people without MetS. The presence of frailty was the outcome, following the PICO framework.

### 2.2. Eligibility Criteria

Only articles written and published in English were considered in this review. Studies were included if they met the following criteria: (1) utilising a standardised and internationally recognised classification framework for the definition of metabolic syndrome (having at least three out of the five components of abdominal obesity, high fasting blood glucose, high blood pressure, high triglyceride level, low high-density lipoprotein level), (2) utilising validated criteria to define frailty, (3) exclusively involving the older population (aged 60 years or older). Exclusion criteria included: (1) studies in which MetS was not treated as a binary variable (yes/no), (2) studies in which frailty was not treated as a binary variable (frail/non-frail), (3) abstract-only papers (conference, editorial, author response) or articles without full text available.

### 2.3. Method of Review

After the initial search, the titles and abstracts of the articles were screened by two independent reviewers for eligibility. Shortlisted articles were downloaded, read in full and excluded should eligibility criteria were not met. Data from the eligible studies were then extracted and inserted in a tabulated manner. The main categories of extracted data included: location, year, design, population characteristics (including sample size, age range of participants), the prevalence of MetS, the prevalence of frailty (in people with and without MetS) and classification frameworks used for definitions of MetS and frailty. After data extraction, quality assessment was independently performed by the aforementioned reviewers, utilising the National Institutes of Health (NIH) Quality Assessment Tool for Observational Cohort and Cross-Sectional Studies. This quality assessment tool comprised 14 assessment items in each study design. Items were scored and studies’ quality was classified as “good,” “fair,” or “poor.” Any disputes were reconciled by a third independent investigator.

### 2.4. Statistical Analysis

To conduct a meta-analysis of the involved data, Review Manager (RevMan), version 5.3 was used. Although classification frameworks for the definition of MetS and frailty within the included studies were standardised and recognised internationally, these frameworks were not exactly uniform. This factor, coupled with varying population sizes, may lead to significant heterogeneity between studies. Thus, random-effects models were deemed appropriate and consequently performed for data pooling. Outcomes were reported as Odd Ratios (OR) and 95% Confidence Interval (CI). Statistical heterogeneity amongst the studies was measured and quantified using Chi-squared (χ^2^) test and I^2^ statistic respectively. The heterogeneity was reported using the corresponding I^2^ thresholds: <25% representing low inconsistency; 50% representing moderate inconsistency and >75% representing high inconsistency.

### 2.5. Sensitivity Analysis

Sensitivity analysis was conducted to determine the effect of the study design, population and study quality (based on the NIH quality assessment tool) on the findings of the meta-analysis (Figure A1, Figure A2, Figure A3, Figure A4, Figure A5 and Figure A6). Based on this information, sensitivity analysis was conducted on the following study parameters:(1)Cross-sectional studies only;(2)Longitudinal studies only;(3)Caucasian studies only;(4)Asian studies only;(5)High-quality studies (those rated as ‘good’ on the NIH quality assessment tool) vs. lower quality studies (those rated ‘fair’ on the NIH quality assessment tool).

## 3. Results

### 3.1. Study Selection and Characteristics

The initial literature search through MEDLINE and EMBASE yielded a total of 798 articles. Amongst those, 203 articles were removed due to duplication, leaving 595 articles to progress onto the next phase for the title and abstract screening. During this process, a further 571 articles were excluded. The remaining 24 articles were reviewed in their full-text forms for eligibility. After removing articles due to the inclusion of younger participants outside the predetermined age limits of our review, lack of classification framework for MetS, or no information of data regarding the prevalence of frailty in people with and without MetS, 11 full-text articles met the eligibility criteria (Figure 1). The 11 articles were subsequently included in our meta-analysis.

Nine of the included studies were cross-sectional [17,18,19,20,21,22,23,24,25], except two studies by Perez-Tasigchana [26] and Barzilay [27], which were longitudinal, reporting a relationship between MetS and frailty at 3.5 years and 9 years after initial recruitment of participants. Additionally, 10 out of 11 studies were conducted in a community-dwelling setting, while one study exclusively considered participants in a clinical setting (outpatients of a geriatric clinic) [17]. The 11 studies all took place in different countries, with 7 studies being conducted in Caucasian populations [17,18,19,21,25,26,27], and 4 studies in Asian populations [20,22,23,24].

With regards to prevalence, the prevalence of frailty ranged from 0.9% to 14.8% in population-based studies and 35.6% in the outpatient clinic setting. The prevalence of MetS was also higher in the outpatient clinic setting at 47.5%, compared to 17.5–41.0% in the community-dwelling populations (Table 1).

### 3.2. The Utilisation of Different Frameworks in Classifying Frailty

Fried’s phenotypic criteria was used to define frailty in 8 out of the 11 included studies [17,18,19,21,22,25,26,27]. According to this classification framework, the presence of at least three out of the five, previously mentioned, adverse-health features (namely, unexplained weight loss, exhaustion, reductions in physical activity, grip strength and gait speed) is required to define frailty. One study utilised the “Study of Osteoporotic Fractures (SOF) Index” as a measure of frailty, which considered only three adverse features: unintentional weight loss, and inability to stand up from a sitting position and fatigue [20]; one study used the FRAIL scale [23], and one study used the Frailty Index [24].

### 3.3. Meta-Analyses

Eleven studies (*n* = 19,270, age > 60) were included for meta-analysis. After data pooling, the pooled estimates revealed that MetS was associated with a higher risk of frailty (OR 1.73, 95% CI, 1.41–2.13). There was moderate heterogeneity between studies (I^2^ = 62%*,*
*p* = 0.003) (Figure 2).

### 3.4. Sensitivity Analysis

Sensitivity analysis showed the largest variations between populations consisting of Asian vs. those consisting of Caucasians with ORs of 1.53 and 1.96 respectively which is consistent with current literature findings within these populations. There was a statistically significant overall effect in all subgroups regardless of population, quality or study design. We were unable to assess the overall effect of confounding variables due to the lack of adjustment within the reported studies (Appendix A
Figure A1, Figure A2, Figure A3, Figure A4, Figure A5 and Figure A6).

### 3.5. Risk of Bias Analysis

The quality of cross-sectional and longitudinal studies included in this review was evaluated in reference to the National Institutes of Health Quality Assessment Tool for Observational Cohort and Cross-Sectional Studies. Of the studies included, six received a rating of ‘good’, five received ‘fair’ and there were no articles included that received a ‘bad’ rating. Of those studies given a ‘fair’ rating score, losses were due to: inadequate sample size justification; lack of measurement of exposure prior to outcomes; adequate timeframe from exposure to outcome; exposure assessed on more than one occasion, and; blinding of the assessors of the outcomes (Appendix A
Table A1).

## 4. Discussion

The results of the meta-analysis of 19,270 participants across 11 different studies showed that older people with MetS were more likely to experience frailty compared to older people without MetS.

The relationship between MetS and frailty may be bidirectional and share the common pathway of chronic low-grade inflammation [1]. MetS can increase the risk of frailty through several mechanisms. MetS is associated with insulin resistance, chronic inflammation, activation of oxidative and prothrombotic pathways, and deregulation of the renin-angiotensin axis, which may have negative impact on various physiological domains that contribute to the developing of frailty over time [17]. MetS, including obesity, leads to increased mobility limitation and, hence, increases the risk of developing sarcopenia and physical frailty. People with MetS are also at high risk of developing cardiovascular diseases (CVD), and there has been evidence that subclinical CVD was strongly associated with incident frailty in community-dwelling population without known CVD [28]. Conversely, frailty can also increase the risk of developing MetS. Frailty is associated with a chronic state of low-grade inflammation, which plays an important role in the developing of MetS. Frail individuals almost always have coexisting sarcopenia, a progressive loss of muscle mass [29]. Sarcopenia is a leading cause of decreased physical functioning, which, in turn, can increase the likelihood of progression of insulin resistance to diabetes and further predisposes these individuals to development of cardiovascular disease [30]. Further investigations into the mechanisms underlying frailty and MetS are needed.

Other predisposing factors for frailty that were not examined in these studies include IGF-1 and Vitamin D, the latter having been independently associated with four of the five frailty components [29]. Insulin-like growth factors (IGF) has a documented role in the development of sarcopenia as well as the pro-inflammatory state that is the hypothesised bidirectional link between MetS and frailty.

The positive association between MetS and frailty highlights the importance of managing MetS and the role this can play in preventing frailty and reducing morbidity and mortality. In people who have MetS, frailty should be screened for, as this is a better predictor of mortality than MetS alone [31]. It has been described previously that the presence of MetS in a frail people is associated with a nine-fold increase in major depressive episodes, twenty-fold increase in heart attack risk and a six-fold increase in strokes when compared to an individual with MetS that is not frail [21]. In addition, MetS and frailty were associated with increased functional impairment with at least one activity of daily living (ADL) affected, poorer quality of life and perceived health and polypharmacy [32].

Therefore, frailty screening and personalised management is important in older people with MetS. Frailty is reversible with early diagnosis and prompt management. Current guidelines from the Asia-Pacific suggest identifying frail individuals through the use of validated measurement tools [12]. The primary treatment goals for frail patients consist of a personally tailored physical-exercise program to preserve muscle force and function in addition to improving glucose metabolism and inflammation which aims to address the low energy or ‘exhaustion’ that is present in the frail phenotype [11,33]. It is also important to manage MetS to avoid worsening frailty. Management of MetS takes a two-pronged approach to drug therapy (to target the modifiable risk factors such as hypertension, hyperglycaemia and dyslipidaemia) and lifestyle intervention (e.g., healthy lifestyle promotion encompassing diet, physical activity, emotional regulation and self-care). Studies have shown that regular exercise in conjunction with health education significantly reduced most parameters of MetS. Older people are more likely to live a sedentary lifestyle due to age-related mobility impairments, so more effort needs to be made to raise awareness about exercise in older adults.

Strength and limitation: 

This is the first systematic review and meta-analysis on the association between metabolic syndrome and frailty. However, this study has several limitations. The studies used in the meta-analysis were primarily cross-sectional studies, which are unable to demonstrate a cause-effect relationship because temporality is not known. The meta-analysis included studies where confounding factors such as age, race, education level, nutritional intake and gender were not adjusted to a satisfactory standard, which may result in bias. The method of classification of MetS and frailty was varied in the included studies, which would limit comparability. Only studies written in English were used and only published data were included in the meta-analysis. Further research is required to elucidate directionality and causal pathways for the development of frailty in older people in relation to MetS.

## 5. Conclusions

Our systematic review and meta-analysis suggested that frailty is more prevalent in older people with MetS compared to older people without MetS. The study findings suggest the importance of screening for frailty in older people with MetS and a distinct role of managing MetS in preventing frailty in older people. Early detection of frailty may allow for optimization of treatment, resulting in better health outcomes. Given the aging population and the rise in obesity, frailty and metabolic syndrome, understanding these co-dynamics will likely be of increasing importance in the future.

## Figures and Tables

**Figure 1 geriatrics-07-00076-f001:**
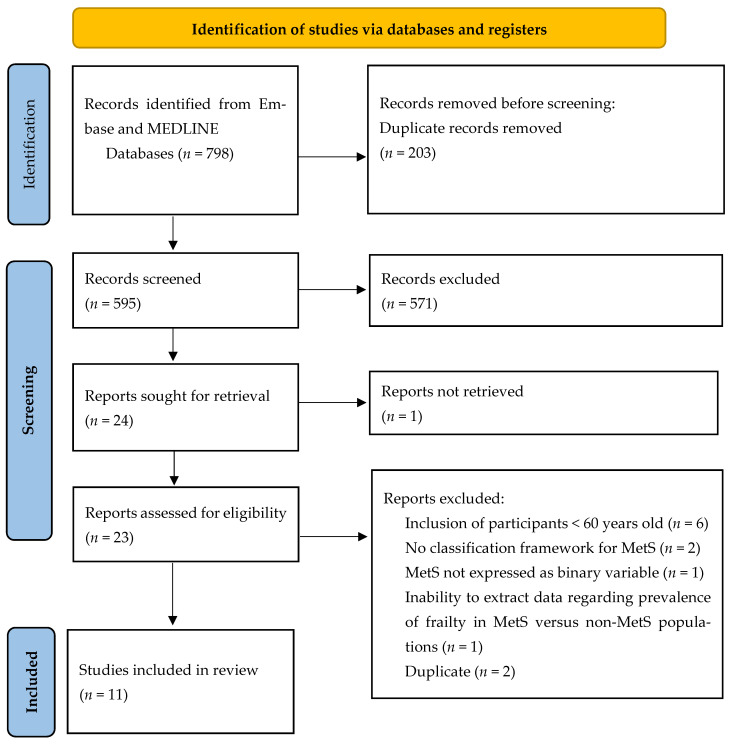
Flowchart of literature search (PRISMA format) detailing the process that resulted in the final study inclusion.

**Figure 2 geriatrics-07-00076-f002:**
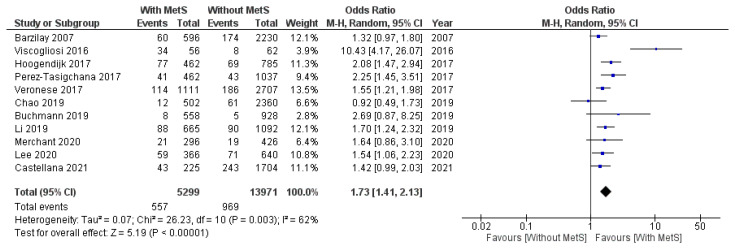
Random effects model meta-analysis of frailty likelihood in people with and without metabolic syndrome in the 11 studies [17,18,19,20,21,22,23,24,25,26,27]. MetS, metabolic syndrome; Events, Frailty.

**Table 1 geriatrics-07-00076-t001:** Characteristics of the included studies.

Author and Year	Country	Population	Size	Definition of Frailty	Prevalence of Frailty	Prevalence of MetS	Method of Classifying MetS	Association
Viscogliosi 2016 [17]	Italy	Mean age 76.1 years Outpatients of geriatric clinics (June–December 2015)	118	Fried’s criteria	Overall: 35.6% (42/118) In participants with MetS: 60.7% (34/56) In participants without MetS: 12.9% (8/62)	Overall: 47.5% (56/118)	National Cholesterol Education Program (NCEP) Adult Treatment Panel III	Cross-sectional
Hoogendijk 2017 [18]	The Netherlands	Mean age 75.4 years Participants of a population based study (The Longitudinal Aging Study Amsterdam)	1247	Fried’s criteria	Overall: 11.7% (146/1247) In participants with MetS: 16.7% (77/462) In participants without MetS: 8.8% (69/785)	Overall: 37.1% (462/1247)	National Cholesterol Education Program (NCEP) Adult Treatment Panel III	Cross-sectional
Perez-Tasigchana 2017 [26]	Spain	Aged ≥ 60 years Participants of a population based study (The Seniors-ENRICA cohort study)	1499	Fried’s criteria	At 3.5 year follow up: Overall: 5.6% (84/1499) In participants with MetS: 8.9% (41/462) In participants without MetS: 4.1% (43/1037)	Overall: 30.8% (462/1499)	Harmonised/joint statement International Diabetes Federation Task Force on Epidemiology and Prevention; National Heart, Lung, and Blood Institute; American Heart Association; World Heart Federation; International Atherosclerosis Society; and International Association for the Study of Obesity in 2009	Longitudinal (prospective cohort study), 3.5-year follow up
Chao 2019 [20]	Taiwan	Mean age 73.4 years Community-dwelling underwent annual health examinations at National Taiwan University Hospital	2862	The Study of Osteoporotic Fractures criteria (SOF)	Overall: 2.6% (73/2862) In participants with MetS: 2.4% (12/502) In participants without MetS: 2.6% (61/2360)	Overall: 17.5% (502/2862)	American Association of Clinical Endocrinologists (AACE)	Cross-sectional
Buchmann 2019 [21]	Germany	Mean age 68.7 years Participants of a population based study (The Berlin Aging Study II)	1486	Fried’s criteria	Overall: 0.9% (13/1486) In participants with MetS: 1.4% (8/558) In participants without MetS: 0.5% (5/928)	37.6% (558/1486)	Harmonised/joint statement International Diabetes Federation Task Force on Epidemiology and Prevention; National Heart, Lung, and Blood Institute; American Heart Association; World Heart Federation; International Atherosclerosis Society; and International Association for the Study of Obesity in 2009	Cross-sectional
Lee 2020 [24]	Taiwan	Mean age 65.8 years Participants of a population based study	1006	Frailty Index (35 items, cut-point to define frailty: FI ≥0.25)	Overall: 12.9% 130/1006 In participants with MetS: 16.1% (59/366) In participants without MetS: 11.1% (71/640)	36.4% (366/1006)	National Cholesterol Education Programme (NCEP) Adult Treatment Panel III (ATP III) guidelines	Cross-sectional
Barzilay 2007 [27]	United States	Aged 69–74 years Participants of the Cardiovascular Health Study	2826	Fried’s criteria	At 9 years: Overall: 8.3% (234/2826) In participants with MetS: 10.1% (60/596) In participants without MetS: 7.8% (174/2230)	21.1% (596/2826)	National Cholesterol Education Programme (NCEP) Adult Treatment Panel III (ATP III) guidelines	Longitudinal/ Prospective cohort study
Veronese 2017 [19]	Iceland	Mean age 76.2 years Participants of a population based Study—The Age, Gene/Environment Susceptibility (AGES)— Reykjavik Study	3818	Fried’s criteria	Overall: 7.9% (300/3818) In participants with MetS: 10.3% (114/1111) In participants without MetS: 6.9% (186/2707)	29.1% (1111/3818)	National Cholesterol Education Programme (NCEP) Adult Treatment Panel III (ATP III) guidelines	Cross-sectional
Li 2019 [22]	China	Mean age 75.3 years Participants of a population based Study (The RuLAS Rugao Longevity and Ageing Study)	1757	Fried’s criteria	Overall: 10.1% (178/1757) In participants with MetS: 13.2% (88/665) In participants without MetS: 8.2% (90/1092)	37.8% (665/1757)	Joint statement between American Heart Association (AHA) and the National Heart, Lung, and Blood Institute (NHLBI) (2005)	Cross-sectional
Merchant 2020 [23]	Singapore	Mean age 71 +/− 5 years Participants of a population based study (The HOPE—Healthy Older People Everyday study)	722	5-point FRAIL scale	Overall: 40/722 = 5.5% In participants with MetS: 7.1% (21/296) In participants without MetS: 4.5% (19/426)	41.0% (296/722)	Modified National Cholesterol Education Programme (NCEP) Adult Treatment Panel III (ATP III) guidelines for Asians	Cross-sectional
Castellana 2021 [25]	Italy	Mean age 73.55 years Participants of a population based Study (the Salus in Apulia study)	1929	Fried’s criteria	Overall: 14.8% (286/1929) In participants with MetS: 19.1% (43/225) In participants without MetS: 14.3% (243/1704)	11.7% (225/1929)	Harmonised/joint statement International Diabetes Federation Task Force on Epidemiology and Prevention; National Heart, Lung, and Blood Institute; American Heart Association; World Heart Federation; International Atherosclerosis Society; and International Association for the Study of Obesity in 2009	Cross-sectional

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
