# Peer review of "The Relationship between Metabolic Syndrome and Frailty in Older People: A Systematic Review and Meta-Analysis"

_geriatrics, 2022, doi:10.3390/geriatrics7040076_

Round 1
Reviewer 1 Report
The topic of the manuscript is very interesting. Indeed, metabolic syndrome is common in frail older adults. However, I have some concerns. The English form should be revised by an English native speaker. Also, please revise the paper for minor spell check. I would suggest you to explain, in the introduction, the role of diabetes in frail older adults; please cite and discuss:
- Frailty and Risk of Fractures in Patients With Type 2 Diabetes.Diabetes Care. 2019 Apr;42(4):507-513. doi: 10.2337/dc18-1965. Epub 2019 Jan 28. PMID: 30692240
- Correlation of physical and cognitive impairment in diabetic and hypertensive frail older adults.PMID: 35045834
The quality of the figure is poor. Please, resubmit the figures in tiff or higher quality.
The discussion should be improved. Please, find some references which could help:
- Clegg A, Hassan-Smith Z.Lancet Diabetes Endocrinol. 2018 Sep;6(9):743-752. doi: 10.1016/S2213-8587(18)30110-4. Epub 2018 Jul 17.PMID: 30017798
- Metabolic Syndrome and Sarcopenia.
Author Response
Thank you for your time reviewing our manuscript and for the useful suggestions. Please see our responses as follows:
The topic of the manuscript is very interesting. Indeed, metabolic syndrome is common in frail older adults. However, I have some concerns. The English form should be revised by an English native speaker. Also, please revise the paper for minor spell check.
Response: Thank you! We have checked and revised the manuscript. Three of the authors of this manuscript are native English speakers (MB, RK, CC).
I would suggest you explain, in the introduction, the role of diabetes in frail older adults; please cite and discuss:
- Frailty and Risk of Fractures in Patients With Type 2 Diabetes.Li G, Prior JC, Leslie WD, Thabane L, Papaioannou A, Josse RG, Kaiser SM, Kovacs CS, Anastassiades T, Towheed T, Davison KS, Levine M, Goltzman D, Adachi JD; CaMos Research Group.Diabetes Care. 2019 Apr;42(4):507-513. doi: 10.2337/dc18-1965. Epub 2019 Jan 28. PMID: 30692240
- Correlation of physical and cognitive impairment in diabetic and hypertensive frail older adults.Mone P, Gambardella J, Lombardi A, Pansini A, De Gennaro S, Leo AL, Famiglietti M, Marro A, Morgante M, Frullone S, De Luca A, Santulli G. Cardiovasc Diabetol. 2022 Jan 19;21(1):10. doi: 10.1186/s12933-021-01442-z.PMID: 35045834
Response: Thank you for the suggestions – we thank you for providing these references. We have added a section to the introduction on the effect of Diabetes on Frailty (lines 65-69).
“Diabetes is known to increase hospitalization, disability, and mortality in frail individuals. Although the full extent of this effect on those with frailty is not known, impairments in physical and cognitive domains have been observed (15). In addition, diabetes and frailty are confounding risk factors for fragility fractures observed at higher rates than expected from these conditions alone(16). “
The quality of the figure is poor. Please, resubmit the figures in tiff or higher quality.
Response: We have revised the figures.
The discussion should be improved. Please, find some references which could help:
- Clegg A, Hassan-Smith Z.Lancet Diabetes Endocrinol. 2018 Sep;6(9):743-752. doi: 10.1016/S2213-8587(18)30110-4. Epub 2018 Jul 17.PMID: 30017798
- Metabolic Syndrome and Sarcopenia. Nishikawa H, Asai A, Fukunishi S, Nishiguchi S, Higuchi K. Nutrients. 2021 Oct 7;13(10):3519. doi: 10.3390/nu13103519. PMID: 34684520
Response: Thank you for your suggestions and the references provided. We have updated our discussion with further factors that influence the relationship between frailty and MetS (please see lines 245-252):
“Frail individuals almost always have coexisting sarcopenia, a progressive loss of muscle mass.(32) Sarcopenia is a leading cause of decreased physical functioning which in turn can increase the likelihood of progression of insulin resistance to diabetes and further predisposes these individuals to development of cardiovascular disease.(33) Other predisposing factors for frailty that were not examined in these studies include IGF-1 and Vitamin D the latter having been independently associated with 4 of the 5 frailty components.(32) Insulin like growth factors (IGF) has a documented role in the development of sarcopenia as well as the pro-inflammatory state that is the hypothesized bidirectional link between MetS and frailty.”
Reviewer 2 Report
This paper should focus on frailty and if MS adds to that condition. The manuscript was suggesting the MS was adding to the frail condition. Its more logical to think that if someone has frailty, is that due in part to them having more of the MS components.
The authors may want to select if any of the 5 MS components were strongly related to frailty condition (HTN or Waist Circum).
I'm not sure Ref 26 should be included (Sarcopenic Obesity is a complex term and may not be understood by the audience)
Author Response
Thank you for your time reviewing our manuscript and for the useful suggestions. Please see our responses as follows:
This paper should focus on frailty and if MS adds to that condition. The manuscript was suggesting the MS was adding to the frail condition. Its more logical to think that if someone has frailty, is that due in part to them having more of the MS components.
Response: Thank you for your suggestion – However given that metabolic syndrome normally develops in middle age which precedes the normal onset of frailty, it is more logical that metabolic syndrome and its constitutive components contribute to the development of frailty in later stages of life.
The authors may want to select if any of the 5 MS components were strongly related to frailty condition (HTN or Waist Circum).
Response: Thank you for your suggestion – unfortunately we cannot examine the relationship between the individual components of MS and frailty using the pooled data. Individual components of MetS have been reported to be linked with frailty by previous publications.
I'm not sure Ref 26 should be included (Sarcopenic Obesity is a complex term and may not be understood by the audience)
Response: Thank you for your suggestion. We have removed this reference.
Round 2
Reviewer 1 Report
The authors have improved the manuscript.
Author Response
Thank you for your time reviewing our manuscript. As there is no further suggestions from you, we'll keep the manuscript in its current form.